# Hydrogen-Rich Gas Enhanced Sprint-Interval Performance: Metabolomic Insights into Underlying Mechanisms

**DOI:** 10.3390/nu16142341

**Published:** 2024-07-19

**Authors:** Gengxin Dong, Haiyan Liu, Yunji Chen, Dapeng Bao, Wentao Xu, Junhong Zhou

**Affiliations:** 1School of Sport Medicine and Physical Therapy, Beijing Sport University, Beijing 100084, China; ddgx0419@163.com; 2School of Huangjiu, Zhejiang Industry Polytechnic College, Shaoxing 312000, China; freeair772000@163.com; 3College of Military and Political Basic Education, National University of Defense Technology, Changsha 410072, China; chenyunji666@bsu.edu.cn; 4China Institute of Sport and Health Science, Beijing Sport University, Beijing 100084, China; 5Key Laboratory of Precision Nutrition and Food Quality, Department of Nutrition and Health, China Agricultural University, Beijing 100084, China; xuwentao@cau.edu.cn; 6Hebrew Senior Life Hinda and Arthur Marcus Institute for Aging Research, Harvard Medical School, Boston, MA 02131, USA; junhongzhou@hsl.harvard.edu

**Keywords:** hydrogen-rich gas, metabolomics, sprint-interval training, anaerobic exercise, fat oxidation

## Abstract

(1) Background: The diversity of blood biomarkers used to assess the metabolic mechanisms of hydrogen limits a comprehensive understanding of its effects on improving exercise performance. This study evaluated the impact of hydrogen-rich gas (HRG) on metabolites following sprint-interval exercise using metabolomics approaches, aiming to elucidate its underlying mechanisms of action. (2) Methods: Ten healthy adult males participated in the Wingate Sprint-interval test (SIT) following 60 min of HRG or placebo (air) inhalation. Venous blood samples were collected for metabolomic analysis both before and after gas inhalation and subsequent to completing the SIT. (3) Results: Compared with the placebo, HRG inhalation significantly improved mean power, fatigue index, and time to peak for the fourth sprint and significantly reduced the attenuation values of peak power, mean power, and time to peak between the first and fourth. Metabolomic analysis highlighted the significant upregulation of acetylcarnitine, propionyl-L-carnitine, hypoxanthine, and xanthine upon HRG inhalation, with enrichment pathway analysis suggesting that HRG may foster fat mobilization by enhancing coenzyme A synthesis, promoting glycerophospholipid metabolism, and suppressing insulin levels. (4) Conclusions: Inhaling HRG before an SIT enhances end-stage anaerobic sprint capabilities and mitigates fatigue. Metabolomic analysis suggests that HRG may enhance ATP recovery during interval stages by accelerating fat oxidation, providing increased energy replenishment for late-stage sprints.

## 1. Introduction

High-intensity exercise causes a large increase in reactive oxygen species (ROS), surpassing the body’s scavenging capacity and breaking the oxidation–antioxidant balance thus inducing oxidative stress [1,2]. The exceeded ROS may also affect mitochondrial function [3], induce decrease in electron transfer and ATP synthesis, diminish the efficiency of aerobic pathways, increase the levels of inorganic phosphate and lactate [4], and disrupt one or more proteins involved in excitation–contraction coupling, leading to reduced muscle force production [5]. These bio-physiological alterations of oxidative stress are one of the major causes of fatigue [5] and consequently affect exercise performance [6]. Therefore, it is highly demanded to reduce excess ROS and thus improve exercise performance.

Hydrogen has the property of selectively scavenging free radicals that are harmful to the body [7]. Studies have shown the benefits of using hydrogen-rich water and hydrogen-rich gas (HRG) for ameliorating oxidative stress and exercise-induced fatigue [8,9,10,11]. However, existing studies assessing mechanisms of hydrogen vary in their selection of blood biomarkers and most markers are not associated with hydrogen effects [9,10], lacking comprehensive screening to identify target compounds associated with hydrogen mechanisms of action. To date, no studies have revealed the complete metabolite changes associated with HRG during exercise and its correlation with physical performance, leaving the mechanisms by which HRG potentially improves exercise performance still widely unexplored.

Metabolomic technology can capture high-throughput snapshots of an organism’s entire metabolic status [12,13,14], providing comprehensive insights into the mechanism underneath the regulation of functional performance by analyzing changes in endogenous metabolites during exercise [15]. Metabolomics has also been widely used to measure the changes in metabolites as induced by the administration of supplements and drugs [16,17]. However, the metabolomic mechanisms through which hydrogen exerts beneficial effects on exercise performance have not been explicitly and comprehensively explored.

Therefore, here, we aimed to explore the identifiable metabolites and metabolic pathways through which HRG can enhance exercise performance and diminish fatigue by using metabolomic technology. We used sprint-interval exercise to induce oxidative stress, a protocol that can provoke comparable or greater systemic redox state responses to high-intensity interval exercise or continuous moderate-intensity exercise, despite a lower total workload [18]. The knowledge obtained by this work can advance the understanding of the mechanisms underlying the benefits of hydrogen in improving exercise performance and alleviating fatigue.

## 2. Materials and Methods

### 2.1. Participants

Ten healthy, recreationally active, adult men (23.6 ± 2.2 yr, 179.4 ± 5.9 cm, 76.5 ± 7.3 kg) were recruited for this study. The inclusion criteria included the following: participants had a minimum of 2 h·d^−1^, 5 d·wk^−1^ training habituation (including resistance training) and had the ability to complete high-intensity cycling ergometer exercises (ability to complete a Wingate test during the preliminary screening visit). The exclusion criteria included the following: no history of lower extremity injury in the six months prior to the experiment, no cardiovascular, respiratory, and endocrine disease. After a detailed instruction of the experimental procedure, each participant signed an informed consent form. The study protocol conformed to the Declaration of Helsinki and was approved by the Exercise Science Experiment Ethics Committee of Beijing Sport University (BSU) (approval number: 2021107H).

### 2.2. Protocol

In this double-blind, counterbalanced, randomized, and crossover-designed study, participants completed two study visits in the laboratory at BSU, during which they completed a sprint-interval test (SIT) using a Wingate power ergometer (Monark 894E, Monark Exercise AB, Vansbro, Sweden) following a 60 min inhalation of HRG (this visit to inhale HRG was recorded as HG) or placebo gas (standard air) (this visit to inhale placebo gas was recorded as PG). The order of the study visits was randomized. Specifically, visit order was randomly allocated according to balanced permutations generated by a web-based computer program (www.randomizer.org (accessed on 10 August 2021)). A minimum washout period of 7 days was provided between these two visits. 

On each of these two visits, for metabolomic analysis, antecubital venous blood samples were collected from participants at three times—before (pre-gas) and immediately after (post-gas) gas inhalation, and immediately after SIT (post-SIT) (Figure 1). All participants were instructed to avoid vigorous exercise, alcohol, coffee, supplements, medicines, and any specific recovery treatments within 48 h of each trial period. On the morning of each study visit, all participants consumed a standardized meal provided by the researchers, consisting of milk, bread, and ham sausage. Furthermore, they abstained from consuming any additional food or beverages prior to the visit.

### 2.3. Hydrogen-Rich Gas (HRG)

HRG was prepared by a hydrogen gas generator, Hydrogen–Oxygen Convalescent Machine 2.5 (Zhiheng Hydrogen Health Technology Co., Ltd., Fuzhou, China). The generator could generate 30 mL/s of hydrogen–oxygen mixed gas (the composition ratio of hydrogen and oxygen was 2:1). HRG was supplied through a nasal cannula attached to a gas generator. Although we could not measure directly the concentrations of hydrogen and oxygen entering the body due to technical limitations, it was mathematically estimated that the average inspiratory flow rate of a healthy young male at quiet would be about 500 mL/s, which far exceeds the flow rate of the hydrogen gas generator, diluting the concentration of inhaled hydrogen, such that the maximum concentration of hydrogen in the inhaled body would be about 4.08%. Similarly, the oxygen concentration would at most be 21.66% [19]. Compared to the oxygen content of the air, the increased oxygen was extremely small. Placebo gas (ambient air, 0.00005% hydrogen, 20.9% oxygen) was supplied by a nasal cannula that was connected to a hydrogen gas generator that did not initiate the hydrogen production program.

### 2.4. Sprint-Interval Test (SIT)

The SIT was used to examine the effect of HRG on sprint-interval performance (a form of exercise to disturb the body’s redox equilibrium). Participants were required to complete four 30 s full-strength sprint rides at the fastest speed, called the first sprint (S1), the second sprint (S2), the third sprint (S3), and the fourth sprint (S4), respectively, with an interval of 4.5 min between each (Figure 1). The participants were required to warm up for 5 min before the formal test. The participants pedaled at their full strength, while the resistance was increased to reach the prescribed load (0.075 × body weight (kg)) within 3 s. During the test, the participants were given verbal encouragement. In addition, participants were asked to avoid lifting their hips off the bike seat during the test. Previous research has demonstrated that this approach effectively disturbs the systemic redox status [18]. Mean Power (MP) was the primary outcome, reflecting the overall performance during a sprint ride. Additionally, indices such as Peak Power (PP), Fatigue Index (FI, (Maximum power−minimum power)/Maximum power × 100%) and Time to Peak (TTP) of each sprint ride were also recorded. Considering that hydrogen does not act until there is a large increase in reactive oxygen species (ROS) after high-intensity exercise, we assessed its impact by the results of S4 (post-hydrogen effect) and the attenuation value (ΔS4-S1) between the S1 and the S4. The S4 results indicate anaerobic performance following high-intensity exercise, while ΔS4-S1 results reflect fatigue and recovery capabilities during sprint-interval exercise.

### 2.5. Hematological Metabolomics

#### 2.5.1. Collection and Processing of Blood Samples

Approximately 5 mL of blood was collected from the participant’s anterior elbow vein and immediately subjected to centrifugation at 1500× *g* for 10 min at 4 °C to separate the plasma. The plasma was then carefully transferred to a new tube and snap-frozen in liquid nitrogen, followed by storage at −80 °C pending assay analysis. Prior to testing, 200 µL of plasma was vortexed with 800 µL of chromatographically pure methanol in a 1.5 mL centrifuge tube for 60 s and then cooled at −20 °C for 1 h. Then, the sample was centrifuged at 15,000 rpm at 4 °C for 10 min. A total of 900 µL of supernatant was extracted and centrifuged under vacuum at 4 °C until dry. Following that, 90 µL of chromatographically pure methanol containing a deuterated internal standard was added, and the mixture was vortexed for 1 min, followed by the addition of 90 µL of ultrapure water and another round of vortexing for 1 min. The samples were centrifuged at 15,000 rpm for 10 min at 4 °C, and the supernatant was then transferred to a new tube for measurement. Finally, 5 µL of supernatant was removed and placed in a 1.5 mL centrifuge tube and vortexed for 60 s for quality control (QC).

To comprehensively capture metabolite information, data acquisition was conducted using both positive and negative ion modes, employing Time-of-Flight Scanning (TOF-SCAN) as the primary collection method. Additionally, the QC samples would also be analyzed using these ion modes, complemented by iterative collection via Automated Tandem Mass Spectrometry (Auto-MS/MS). The use of TOF-SCAN was intended to monitor the stability of the instrument throughout the sample processing, while Auto-MS/MS was implemented to obtain detailed fragment ion information of the compounds, which is crucial for accurate qualitative confirmation during data analysis. In the positive ion mode, a C18 column (2.1 × 150 mm, 3.5 μm) was utilized, with the column temperature set at 50 °C. The mobile phase A consisted of 0.1% formic acid in water, while mobile phase B comprised 0.1% formic acid in acetonitrile. The flow rate was maintained at 0.35 mL/min. For the negative ion mode, a T3 column (2.1 × 100 mm, 1.8 μm) was employed, also at a column temperature of 50 °C. Here, mobile phase A included 6.5 mmol/L ammonium bicarbonate in water, and mobile phase B was a mixture of 6.5 mmol/L ammonium bicarbonate in 95% methanol–water. The flow rate was consistently set at 0.35 mL/min for this mode as well. These parameters were carefully chosen to optimize the separation and detection of metabolites in each ionization mode.

#### 2.5.2. Non-Targeted Metabolomics

In our metabolomics study, we initially processed the raw metabolite data, detected in full-scan mode, using MassHunter B.06.00 for Molecular Feature Extraction (MFE) and then exported the data in CEF format. We then imported this data into Mass Profiler Professional (MPP) 14.9.1 software. Within MPP, we aligned the data by setting parameters including adduct ion types and modes, isotope types, compound charge states, retention times, mass range, mass filter parameters, signal-to-noise ratio, and minimum peak height. To improve data analysis quality, we re-identified the ions under specific recursive conditions.

In response to the complex nature of the metabolomics data, characterized by high dimensionality and significant inter-variable correlation, traditional univariate analysis methods were deemed inadequate for efficient and accurate information extraction. Therefore, we utilized multivariate statistical techniques, specifically Principal Component Analysis (PCA), to reduce the dimensionality of the data. PCA was applied to transform the large set of variables into a smaller set that retains most of the original data’s variance, facilitating subsequent regression analysis on this reduced dataset. 

For all blood metabolites, we sequentially filtered differential compounds using MPP software based on compound marking (Filter Flags) and occurrence frequency (Filter By Frequency). We utilized PCA to assess sample quality and explore differences in blood metabolites. Positive ion mode filtering conditions were as follows: frequency > 50%, response value filtering > 10,000, *p* < 0.05, fold change thresholds (FC) ≥ 1.5; Negative ion mode screening conditions were as follows: frequency > 50%, response value filtering > 7000, *p* < 0.05, FC ≥ 1.5. Subsequently, using the Metlin database—which contains information on over 80,000 metabolites and secondary mass spectrometry data for approximately 11,000—we identified and analyzed differential compounds to determine the identity and structural features of the metabolites, laying a foundation for biological interpretation.

#### 2.5.3. Targeted Metabolomics

Initial identification of target compounds included a review of the literature on metabolites associated with high-intensity exercise. This was followed by mass spectrometry analysis to screen for compounds. However, some were excluded due to inadequate responses in first-level screening or insufficient quality of second-level mass spectra. Ultimately, 15 compounds met the criteria and were successfully incorporated into the Personal Compound Database and Library (PCDL), as detailed in Table 1. The prepared samples were tested on the machine, and the test data were qualitatively matched with the established PCDL to analyze whether the target compounds were present in the samples and if there were response value differences in the detected compounds. These 15 metabolites were then used as mixed standard solutions at a concentration of 1 μg/mL for controlled quantitative analysis. These 15 amino acid metabolites were ultimately targeted using MassHunter quantitative software. The mass spectral fragment ion information of these metabolites is displayed in Table 1.

### 2.6. Statistical Analysis

Statistical analyses were performed using SPSS 25.0 (IBM, Chicago, IL, USA). A value of *p* < 0.05 was considered statistically significant. Descriptive statistics (i.e., mean, standard deviation (SD)) were used to summarize the demographic characteristics of the participants. Shapiro–Wilk tests were used to examine if the data were normally distributed. Data that were normally distributed were described using “mean ± SD”. The performance results of SIT were analyzed for differences by paired *t*-test models, including differences in MP, PP, FI, and TTP of S4, and the difference in the attenuation values of these parameters (i.e., the ΔS4–S1 value) between HG and PG, to elucidate the influence of HRG on SIT. Metabolomics results were analyzed by MPP software and combined with statistical models, setting the significance level at *p* < 0.05 and a fold change threshold (FC ≥ 1.5) to distinguish differential metabolites. Analysis of variance (ANOVA) models were used to test for the differences in metabolites between the three time points (pre-gas, post-gas, and post-SIT) in the HG and PG. The *t*-test models were then used to examine the differences in these differential metabolites between HG and PG, specifically focusing on the differences in metabolic concentration changes post-gas and post-SIT between the HG and PG to assess the effect of HRG on the metabolic profile of individuals after SIT. The screened differential metabolites were matched against the Kyoto Encyclopedia of Genes and Genomes (KEGG) database to enrich the relevant pathways.

## 3. Results

All ten participants completed all study tests, and their data were included in the analysis. None of the subjects were able to distinguish the difference between hydrogen and placebo gas. They also reported no side effects with the hydrogen inhalation.

### 3.1. Inhalation of HRG Induced Improvements in Sprint-Interval Test

The primary paired *t*-test model showed that the MP of S4 in HG was significantly higher than that in PG (t = 2.585, *p* = 0.029), and the MP of ΔS4-S1 in HG were significantly greater than that in PG (t = 3.579, *p* = 0.006). Secondarily, the FI and TTP of S4 in HG was significantly lower than that in PG (FI: t = −4.567, *p* = 0.001; TTP: t = −3.643, *p* = 0.005), but the PP of S4 was not significantly different in HG and PG (t = 2.114, *p* = 0.064) (Higher PP of S4 was observed after inhaling HRG compared to placebo; however, the difference was not statistically significant. This may be attributed to insufficient statistical power resulting from the relatively small sample size.). The PP of ΔS4-S1 in HG were significantly greater than that in PG (t = 5.046, *p* = 0.001), the TTP of ΔS4-S1 in HG was significantly smaller than that in PG (t = −3.939, *p* = 0.003), but the FI of ΔS4-S1 was not significantly different in HG and PG (t = 0.234, *p* = 0.820) (Table 2). These results suggested that pre-exercise inhalation of HRG helped the maintenance of anaerobic performance during sprint ride after high-intensity exercise and mitigates fatigue.

### 3.2. Non-Targeted Metabolomics Test Results

The MPP software identified a total of 5717 metabolites from the data acquired in positive-ion mode. Filtering the data for metabolites with a frequency of occurrence over 50% yielded 1639 metabolites. Further refinement based on response values resulted in 474 metabolites. After excluding compounds originating from solvents, a final count of 300 metabolites remained. In HG and PG, there was no significant difference in each metabolite before and after inhalation (*p* > 0.05), but 18 metabolites showed significant differences before and after SIT (*p* < 0.05). After comparison with the METLIN database, four metabolites with significant differences were identified. Further analysis of these four metabolites’ ΔSIT between HG and PG revealed that acetylcarnitine, PS(O-18:0/0:0), and propionyl-L-carnitine were significantly more upregulated in HG compared to PG, while 4-methylphenyl octanate was significantly more downregulated in HG than in PG (*p* < 0.05).

The MPP software identified a total of 11,683 metabolites from the data acquired in negative-ion mode. Filtering the data for metabolites with a frequency of occurrence over 50% yielded 3903 metabolites. Further refinement based on response values resulted in 373 metabolites, none of which were compounds originating from solvents. In HG and PG, there was no significant difference in each metabolite before and after inhalation (*p* > 0.05), but 14 metabolites showed significant differences before and after the SIT (*p* < 0.05). After comparison with the METLIN database, four metabolites with significant differences were identified. Coriose and methoxyacetic acid were significantly upregulated after the SIT (*p* < 0.05), but their ΔSIT were not significantly different between HG and PG (*p* > 0.05). (±)-6-methyl-caprylic acid and 2-(Ethylsulfinylmethyl) phenyl methylcarbamate was significantly downregulated after SIT (*p* < 0.05), and 2-(Ethylsulfinylmethyl) phenyl methylcarbamate was significantly more downregulated in HG than in PG (*p* < 0.05). However, (±)-6-methyl-caprylic’ ΔSIT was not significantly different between HG and PG (*p* > 0.05). 

The secondary data iteratively collected from the samples were subjected to secondary mass spectrometry confirmation using MassHunter qualitative software, and of the above non-targeted screened metabolites, acetylcarnitine and propionyl-L-carnitine were characterized with at least one fragmentation ion in addition to their molecular ion peaks, facilitating their qualitative identification.

### 3.3. Targeted Metabolomics Test Results

The 15 amino acid metabolites in all samples were targeted using MassHunter quantitative software. In HG and PG, there was no significant difference in each metabolite before and after inhalation (*p* > 0.05), but three metabolites (hypoxanthine, xanthine, D-pantothenic acid) showed significant differences before and after the SIT (*p* < 0.05). The data collected for the targeting analysis were analyzed by the MPP software. Hypoxanthine was upregulated after SIT in positive ion mode, and the upregulation of HG was significantly greater than that of PG (*p* < 0.05). Hypoxanthine and xanthine were upregulated after SIT in negative-ion mode, and the upregulation of HG was significantly greater than that of PG (*p* < 0.05).

### 3.4. Multiple Metabolic Pathways Associated with Differential Metabolites

Metabolic pathway analysis was performed on the differential metabolites in HG, and KEGG database matching analysis showed the top 20 significantly enriched relevant pathways. Among these, certain molecular characteristics were related to hydrogen absorption metabolism, such as glycerophospholipid metabolism, pantothenate, and coenzyme A (CoA) biosynthesis, and insulin resistance (Figure 2).

## 4. Discussion

To our knowledge, this is the first study to explore the potential mechanisms through which pre-exercise inhalation of HRG improves sprint-interval exercise performance using metabolomics analyses. Four types of compounds were found to be the potential key mechanistic metabolites of such benefits of HRG, including acetylcarnitine and propionyl-L-carnitine from non-targeted analysis and hypoxanthine and xanthine in targeted analysis.

Hypoxanthine, a purine derivative found naturally, acts as an inosine in tRNA anticodons and is a nucleic acid component [20]. Plasma hypoxanthine concentration varies with exercise intensity. Hypoxanthine levels rise exponentially following high-intensity exercise [21,22]. High-intensity exercise induces ATP hydrolysis, increasing ADP and hypoxia. This process enhances the metabolism of ADP within the cell. Under hypoxic conditions, one significant pathway of ADP metabolism involves its conversion to adenosine by adenosine kinase, followed by the degradation of adenosine into hypoxanthine by adenosine deaminase. As cellular ATP consumption increases, so does the production of hypoxanthine [23]. Hypoxanthine is subsequently converted to xanthine by xanthine oxidase [24]. In our study, the inhalation of HRG possessed a better performance of the SIT, along with higher concentrations of hypoxanthine and xanthine, suggesting that more ATP is constantly being broken down to provide energy during the sprinting process after hydrogen inhalation. Acetylcarnitine and propionyl-L-carnitine are esters formed when short-chain fatty acids (SCFA) are combined with carnitine, and their primary biological function is to transport fatty acids from the cytoplasm to the mitochondrial matrix for oxidation [25]. Increased fat mobilization correlates with higher SCFA levels. In the results of our study, there were higher concentrations of acetylcarnitine and propionyl-L-carnitine after the inhalation of HRG, suggesting that hydrogen may have enhanced mitochondrial lipid metabolism and accelerated the β-oxidative catabolism of fatty acids [26]. Previous studies have also shown that hydrogen increases fat oxidation [27,28]. 

During the SIT, a large amount of ROS is produced in the body of participants after such high-intensity sprinting exercise [29], which causes oxidative damage to mitochondria and reduces subsequent aerobic utilization [4]. This leads to inadequate ATP replenishment, diminishing subsequent anaerobic sprint performance. Normally, the electron transport chain creates a proton gradient, facilitating ATP production [30]. But ATP can also be produced independently by establishing a hydrogen gradient [31]. Dohi found that hydrogen molecules could create a gradient boosting mitochondrial ATP production independently of the electron transport chain [32], whereas our study found that hydrogen resulted in higher concentrations of acetylcarnitine and propionyl-L-carnitine, suggesting that hydrogen may enhance mitochondrial lipid metabolism, accelerating the β-oxidative catabolism of fatty acids and generating more ATP. Additionally, the higher concentrations of hypoxanthine and xanthine also indicate that more ATP is continually broken down to supply energy. In the SIT, the 30 s sprint phase relies on ATP–CP and anaerobic glycolysis, while the 4.5 min interval involves aerobic oxidation, in which, among others, fat and glucose are the main energy sources to replenish the phosphocreatine reserves and provide the energy required for the next anaerobic sprint. Hydrogen scavenges the large amount of ROS produced during high-intensity exercise and reduces their oxidative damage to the mitochondria, thereby increasing the utilization of aerobic pathways [4,19,33]. Reduced oxidative damage makes mitochondria more efficient at converting substrates to ATP, supplying more cellular energy. Increased mitochondrial efficiency does not directly impact the ATP–CP system’s function, but contributes to aerobic oxidation during exercise intervals, which utilizes oxygen to re-synthesize phosphocreatine to provide more energy for the next exercise. This may be one reason for the improved S4 performance and reduced fatigue (ΔS4–S1). The direct link between hydrogen’s impact on ROS and muscle metabolism was not definitively established in our data, representing a limitation of our study. We conducted preliminary explorations based on existing evidence and our observations, which serve to lay a foundation and provide inspiration for future studies. 

Pantothenic acid, a precursor to CoA, is crucial for CoA-dependent reactions, notably in lipid metabolism and fatty acid β-oxidation [34]. Pathway enrichment analyses have demonstrated a strengthening in the biosynthesis pathways of pantothenic acid and CoA, suggesting that hydrogen could potentially enhance CoA synthesis, thereby facilitating the production of acylcarnitine and augmenting lipid oxidation. During glycerophospholipid metabolism, the breakdown of glycerophospholipids into free fatty acids aids the lipid oxidation process. The enrichment observed in glycerophospholipid metabolism, as well as in the pathways of fat digestion and absorption, emphasizes hydrogen’s role in promoting lipid metabolism. Additionally, the hormonal regulation of enzymes involved in fat mobilization, especially the impact of insulin on the activity of hormone-sensitive lipase, illustrates the complex interplay between fat mobilization and insulin sensitivity, which is known to improve with physical exercise [35]. However, hydrogen’s capacity to suppress insulin level increases may activate hormone-sensitive lipase, thereby enhancing lipolysis [36,37]. This concurs with the enrichment seen in the insulin resistance metabolic pathway. Nonetheless, this suppression might affect glucose uptake, and whether this may adversely affect prolonged repetitive sprinting exercise needs to be further explored. This is essential to provide a more comprehensive insight into the potential mechanisms by which hydrogen affects metabolic processes.

## 5. Conclusions

In conclusion, inhalation of HRG prior to SIT improves the final anaerobic sprint performance and mitigates fatigue. Metabolomic analysis suggests that the action mechanism of HRG may involve mitigating oxidative damage to mitochondria by scavenging the substantial amounts of ROS generated during initial sprint activities. This process enhances the efficiency of aerobic energy utilization. Furthermore, HRG may boost fat oxidation through increased synthesis of coenzyme A, enhancement of glycerophospholipid metabolism, and reduction of insulin levels. These effects collectively support rapid replenishment of ATP during intermittent periods, thereby providing augmented energy reserves for subsequent sprint phases.

## Figures and Tables

**Figure 1 nutrients-16-02341-f001:**
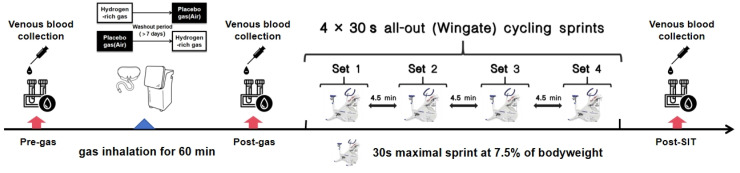
Experimental test sequence and procedure. Pre-gas: before gas inhalation; post-gas: immediately after gas inhalation; Post-SIT: immediately after sprint-interval test.

**Figure 2 nutrients-16-02341-f002:**
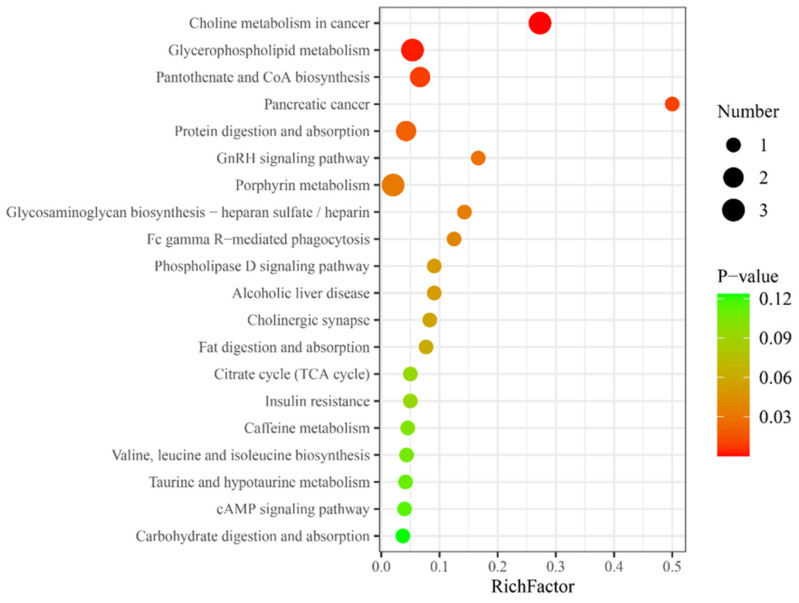
Top 20 differential metabolic pathways for differential metabolites in the hydrogen group. RichFactor represents the ratio of the number of differential metabolites in a metabolic pathway to the total number of metabolites identified in that pathway, a higher value indicates a higher degree of enrichment of differential metabolites within the pathway; the *y*-axis represents the differential metabolic pathways, such as choline metabolism in cancer, which suggests that hydrogen-rich gas, compared to air, can promote the choline metabolic pathway in cancer; Number, the size of the dot represents the number of differential metabolites in the corresponding pathway, a larger dot indicates a greater number of differential metabolites within that pathway; *p*-value, represents the *p*-value from the hypergeometric test, a smaller *p*-value indicates greater reliability and statistical significance of the test.

**Table 1 nutrients-16-02341-t001:** Targeted metabolite information.

Metabolite Name	Molecular Formula	Stock Solution Concentration (mg/L)	Volumetric Solution	Ion Modes	Mass Spectrometry Ion Fragments
Creatine	C_4_H_9_N_3_O_2_	587.0	90% methanol	Pos	132.0768, 90.0550, 72.0556, 57.0447, 58.0526
				Neg	130.0622, 88.0404
Creatinine	C_4_H_7_N_3_O	543.0	90% methanol	Pos	114.0662, 44.0495, 43.0291, 86.0713, 45.0520
				Neg	112.0516, 41.0145, 43.0064
L(-)-Carnitine	C_7_H_15_NO_3_	1000.0	/	Pos	162.1125, 60.0808, 103.0390, 58.0651, 85.0284
				Neg	
D-Carnitine	C_7_H_15_NO_3_	1000.0	/	Pos	162.1125, 60.0808, 103.0390, 58.0651, 85.0284
				Neg	
D-Pantothenic acid	C_9_H_17_NO_5_	550.0	90% methanol	Pos	220.1179
				Neg	
Hydrocortisone	C_21_H_30_O_5_	128.0	90% methanol	Pos	363.2166
				Neg	422.2310, 331.1915, 297.1496, 282.1261, 125.0608
Xanthine	C_5_H_4_N_4_O_2_	146.0	Methanol: 1.5 mol/L NaOH (1:3)	Pos	
				Neg	151.0261, 108.0203, 65.9985, 80.0254, 152.0282
Hypoxanthine	C_5_H_4_N_4_O	554.0	Methanol: 10% formic acid (1:3)	Pos	137.0458
				Neg	135.0312, 92.0254, 65.0145, 64.0067, 66.0098
L-Histidine	C_6_H_9_N_3_O_2_	621.0	90% methanol	Pos	156.0768, 56.0495, 110.0713, 83.0604, 82.0526
				Neg	154.0622, 93.0458, 137.0356, 80.0380, 81.0458
L-Isoleucine	C_6_H_13_NO_2_	1000.0	/	Pos	132.1019, 44.0495, 86.0964, 41.0386, 56.0495
				Neg	130.0874
D-Glutamic acid	C_5_H_9_NO_4_	142.0	90% methanol	Pos	
				Neg	146.0459
L-serine	C_3_H_7_NO_3_	577.0	90% methanol	Pos	
				Neg	104.0353, 74.0247
L-Threonine	C_4_H_9_NO_3_	1432.0	90% methanol	Pos	
				Neg	118.0510, 74.0247
L-Valine	C_5_H_11_NO_2_	730.0	90% methanol	Pos	
				Neg	116.0717
βHydroxybutyric acid	C_4_H_8_O_3_	517.0	90% methanol	Pos	
				Neg	103.0401, 59.0139

Note: Pos, positive ion mode; Neg, negative ion mode.

**Table 2 nutrients-16-02341-t002:** The results of sprint-interval test.

		HG	PG	*p*-Value
S4	MP (W/kg)	6.32 ± 0.74	5.73 ± 0.77	0.029
	PP (W/kg)	8.69 ± 1.19	7.92 ± 1.28	0.064
	FI (%)	65.3 ± 12.3	75.2 ± 14.4	0.001
	TTP (S)	4.72 ± 2.62	9.10 ± 4.00	0.005
ΔS4-S1	MP (W/kg)	−2.02 ± 0.97	−2.77 ± 1.00	0.006
	PP (W/kg)	−2.41 ± 1.58	−3.87 ± 1.84	0.001
	FI (%)	13.2 ± 10.3	11.9 ± 17.1	0.820
	TTP (S)	2.09 ± 2.52	6.82 ± 4.04	0.003

Note: HG, visit with inhalation of hydrogen-rich gas; PG, visit with inhalation of placebo gas; S4, the fourth sprint; ΔS4-S1, the attenuation value between the first sprint and the fourth sprint; MP, mean power; PP, peak power; FI, fatigue index; TTP, time to peak.

## Data Availability

Partial data from this study are included in the original text. For more detailed data, approval from our institution is required due to privacy concerns. Please contact the corresponding author for further inquiries.

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
