# Peer review of "Hydrogen-Rich Gas Enhanced Sprint-Interval Performance: Metabolomic Insights into Underlying Mechanisms"

_nutrients, 2024, doi:10.3390/nu16142341_

Round 1

Reviewer 1 Report

Comments and Suggestions for Authors

The authors have done a great job at assessing the effect of hydrogen-enriched air inhalation on repeated-sprint performance. The paper was interesting to read. The study is well-controlled and has a high degree of novelty. This reviewer only has minor comments that will increase the quality of the paper. Particularly, the discussion and some of the interpretations by the authors are a bit doubtful and some clarity is required here.

Abstract

Line 32: Sure about that it is due to increasing fat metabolism? Fat is not a predominant substrate during sprints, making your conclusion less likely. Please explain

Line 62: Amend sentence/question, as this is not a question, but a question mark is set at the end.

Methods

Methods are well-described. Only some minor comments below.

Study design is randomised, randomisation procedure well-explained.

Placebo well-produced vs. HRG-gas inhalation, although a study limitation is the added O2 content to the inhaled HRG-air.

Line 142-145: Explain the procedure of plasma extraction – when in the process (before snap-freezing in liquid N2?) and how (centrifugation step?)

Line 153: “will be” is future tense, but the study is not in the future

Line 180: describe the exact statistical methods used.

Line 185: “in” blood samples?

Line 194: Study limitation is focusing on metabolites based on literature review of high-intensity exercise effect. What could have been done differently to avoid overlooking important, not previously identified, molecules of interest?

Line 205: you mean “is shown”

Line 213: What about the data that were NOT normally distributed? Did you analyse this using RANK-tests, as is normal procedure for non-normally distributed data.

What data were not normally distributed?

Line 221: It may be more appropriate to use posthoc comparisons tests, such as Tukey’s, and not t-tests. Please justify your choice or if not possible choose an appropriate posthoc test.

Results

Report of well-controlled placebo

Report of t-statistic, accounting for variability and p-values.

Please report sprint data, including MP, PP, TTP, and FI in figures as means with standard deviations or confidence intervals, or another appropriate way of showing variability and comparisons between groups visually.

Figure 2 – In the legend, explain the figure please, including explanation of “Rich Factor”, “Number”, and “P-value”, explain also very briefly in the legend the cancer pathways, for example: HG-air promoted metabolic pathways associated with cancers….vs. normal air inhalation.

Discussion

Line 302: explain how enhancing ADP metabolism may boost hypoxanthine production – it is unclear in the text.

Line 305: I don’t think your point can explain your data/results about xanthine and hypoxanthine. A more likely explanation may be the effect that hydrogen has on scavenging OH· and ONOO-, providing a less favourable environment for XO activity, slowing down these metabolites’ degradation to uric acid. The effect hydrogen has on ROS scavenging could be the key performance boosting mechanism, rather than providing more ATP, as you suggest. What do you think? Please reconsider your interpretation.

Need to explain why delta S1-S4 PP was not different between groups and what this means

Line 324-325: fatty acid oxidation is a slow process. So please explain why this may boost SPRINT performance – this should be much clearer.

Line 325-326: “the higher concentrations of Hypoxanthine and Xanthine also indicating that more ATP is continually broken down …” I do not follow this point. Please explain the relevant pathway/mechanism here.

Line 335: “thereby enhancing…” It is the other way around: boosting ATP replenishment increases ability to recover!

Line 336: “This results in improved S4 performance and reduced fatigue” you do not know this for a fact. Although your data might suggest an association. Amend this sentence.

Line 348-353: May the depressing effect of hydrogen on increases in insulin concentration be bad for repeated sprint performance in the long run when glycogen levels become depleted? Because insulin is important for muscle’s blood glucose uptake. In other words, does hydrogen limit glucose uptake so that less glucose becomes available for ATP production over time, for example in endurance cycling races, where sprints are performed at intervals but a high glycogen-spending intensity is sustained over hours.

Comments on the Quality of English Language

The quality of the English language is generally good, with only minor corrections required.

Author Response

Comment 1: Line 32: Sure about that it is due to increasing fat metabolism? Fat is not a predominant substrate during sprints, making your conclusion less likely. Please explain

Response 1: Thank you for pointing this out. We agree with the idea that fat energy supply is not the primary modality in sprinting. According to the metabolomics results, the changes in metabolites and enrichment pathways suggest that hydrogen increases fat oxidation, which doesn't play a role in sprinting, but rather has to be quickly recovered from energy substances during intermittent rest to provide more energy for the next sprint. Therefore, we modified the description of the sentence, in line 32.

Comment 2: Line 62: Amend sentence/question, as this is not a question, but a question mark is set at the end.

Response 2: Thank you for pointing this out. We have changed the sentence to a declarative sentence, in line 64.

Comment 3: Line 142-145: Explain the procedure of plasma extraction – when in the process (before snap-freezing in liquid N2?) and how (centrifugation step?)

Response 3: Thanks. We have described the information on plasma extraction in detail in line 148.

Comment 4: Line 153: “will be” is future tense, but the study is not in the future

Response 4: Thanks. Changed to past tense.

Comment 5: Line 180: describe the exact statistical methods used.

Response 5: Thanks. Principal component analysis methods have been described.

Comment 6: Line 185: “in” blood samples?

Response 6: Explored the differences in blood metabolites. Modifications have been made.

Comment 7: Line 194: Study limitation is focusing on metabolites based on literature review of high-intensity exercise effect. What could have been done differently to avoid overlooking important, not previously identified, molecules of interest?

Response 7: Agree. We also conducted a non-targeted, comprehensive screening to avoid overlooking compounds not detected in previous studies, as detailed in section 2.5.2.

Comment 8: Line 205: you mean “is shown”

Response 8: We have made modifications.

Comment 9: Line 213: What about the data that were NOT normally distributed? Did you analyse this using RANK-tests, as is normal procedure for non-normally distributed data.

Response 9: Thank you for pointing this out. There is no non-normally distributed data here.

Comment 10: What data were not normally distributed?

Response 10: There is no non-normally distributed data here.

Comment 11: Line 221: It may be more appropriate to use posthoc comparisons tests, such as Tukey’s, and not t-tests. Please justify your choice or if not possible choose an appropriate posthoc test.

Response 11: Thanks. We considered the following: to compare the differences in metabolites between the two groups, only one comparison was used for each set of data, and there were no multiple comparisons, so a statistical model of simple comparisons was chosen for the analyses, which would not have resulted in a Type I error.

Comment 12: Report of well-controlled placebo

Response 12: Thanks. Reports of good control for placebo are described on line 240. Each subject was asked about their feelings and perceptions after the two interventions, and no subject noticed a difference between the two interventions.

Comment 13: Report of t-statistic, accounting for variability and p-values.

Response 13: Thanks. In the main text, the t-statistic results for exercise performance are described. For the metabolomics analysis, metabolites with significant differences were identified by setting a p-value threshold, as explained in the statistical description section. Therefore, the results are presented solely in terms of the relationship between the p-values and the threshold of 0.05.

Comment 14: Please report sprint data, including MP, PP, TTP, and FI in figures as means with standard deviations or confidence intervals, or another appropriate way of showing variability and comparisons between groups visually.

Response 14: Thank you for pointing this out. We have added Table 2 to show these data.

Comment 15: Figure 2 – In the legend, explain the figure please, including explanation of “Rich Factor”, “Number”, and “P-value”, explain also very briefly in the legend the cancer pathways, for example: HG-air promoted metabolic pathways associated with cancers….vs. normal air inhalation.

Response 15: Thank you for pointing this out. We have added this information to the legend.

Comment 16: Line 302: explain how enhancing ADP metabolism may boost hypoxanthine production – it is unclear in the text.

Response 16: Thank you for pointing this out. We have enriched this information in line 329.

Comment 17: Line 305: I don’t think your point can explain your data/results about xanthine and hypoxanthine. A more likely explanation may be the effect that hydrogen has on scavenging OH· and ONOO-, providing a less favourable environment for XO activity, slowing down these metabolites’ degradation to uric acid. The effect hydrogen has on ROS scavenging could be the key performance boosting mechanism, rather than providing more ATP, as you suggest. What do you think? Please reconsider your interpretation.

Response 17: Thanks. We agree that hydrogen removal of reactive oxygen species is key. Considering it further, the removal of reactive oxygen species facilitates the reduction of their damage to mitochondria and improves the aerobic utilisation of mitochondria. This is the most common understanding at present. Improved mitochondrial aerobic function produces more ATP to be consumed, hence the increase in hypoxanthines and xanthines. This is our consideration. We are redescribing how hypoxanthine is produced, which may help in understanding.

Comment 18: Need to explain why delta S1-S4 PP was not different between groups and what this means

Response 18: Higher PP of S4 was observed after inhaling HRG compared to placebo; however, the difference was not statistically significant. This may be attributed to insufficient statistical power resulting from the relatively small sample size. We explained this in line 248.

Comment 19: Line 324-325: fatty acid oxidation is a slow process. So please explain why this may boost SPRINT performance – this should be much clearer.

Response 19: In SIT, the 30-second sprint phase relies on ATP-CP and anaerobic glycolysis, while the 4.5-minute interval involves aerobic oxidation, in which among others, fat and glucose are the main energy sources to replenish phosphocreatine reserves to provide the energy required for the next anaerobic sprint.

Comment 20: Line 325-326: “the higher concentrations of Hypoxanthine and Xanthine also indicating that more ATP is continually broken down …” I do not follow this point. Please explain the relevant pathway/mechanism here.

Response 20: Line 330 explains the relationship between increased hypoxanthine and xanthine and ATP consumption, with the same considerations as Response 17.

Comment 21: Line 335: “thereby enhancing…” It is the other way around: boosting ATP replenishment increases ability to recover!

Response 21: Thank you for pointing this out. We changed the description to “Increased mitochondrial efficiency does not directly impact the ATP-CP system's function, but contributes to aerobic oxidation during exercise intervals, which utilises oxygen to re-synthesise phosphocreatine to provide more energy for the next exercise.”

Comment 22: Line 336: “This results in improved S4 performance and reduced fatigue” you do not know this for a fact. Although your data might suggest an association. Amend this sentence.

Response 21: Thank you for pointing this out. We changed the description to “This may be one reason for the improved S4 performance and reduced fatigue.”

Comment 23: Line 348-353: May the depressing effect of hydrogen on increases in insulin concentration be bad for repeated sprint performance in the long run when glycogen levels become depleted? Because insulin is important for muscle’s blood glucose uptake. In other words, does hydrogen limit glucose uptake so that less glucose becomes available for ATP production over time, for example in endurance cycling races, where sprints are performed at intervals but a high glycogen-spending intensity is sustained over hours.

回应 23:感谢您对氢气对耐力运动中胰岛素抑制的潜在长期影响的有见地的评论。我们认为这是一个重要问题。不幸的是,我们的研究只进行了四次30秒的冲刺,这可能掩盖了胰岛素抑制的负面影响。长时间运动中减少糖摄入的影响需要进一步探讨,我们在手稿中提出了这一点。

感谢您的反馈,这无疑加强了手稿。我们还试图澄清几个问题,希望您能更好地理解我们的观点。如果有任何不清楚的地方,请通知我们,以便我们提供进一步的解释和修改。

Reviewer 2 Report

Comments and Suggestions for Authors

The manuscript lacks a scientific basis and proper support of results:

- It is to be better proved that hydrogen acts as ROS scavenger.

The metabolomic analysis is not correctly linked to ROS metabolism, but rather to muscle catabolism. However, the three pathways are enormously different.

The method should be compared to another ROS-limiting effect, such as supplements, which are more easily handled.

- Even if efficacy, inhaling for 1 hour before training or match is very cumbersome.

Comments on the Quality of English Language

It needs revision.

Author Response

Comment 1: The manuscript lacks a scientific basis and proper support of results:

It is to be better proved that hydrogen acts as ROS scavenger.

The metabolomic analysis is not correctly linked to ROS metabolism, but rather to muscle catabolism. However, the three pathways are enormously different.

Response 1: Thank you for pointing this out. In our previous research, we demonstrated that hydrogen eliminates reactive oxygen species (ROS) and reduces oxidative stress, supported by substantial evidence regarding the relationship between hydrogen and ROS. To further explore the underlying mechanisms of hydrogen, we conducted a metabolomics analysis. Indeed, a direct link between hydrogen, ROS, and muscle metabolism is not explicitly established in our data; this constitutes a limitation of our study. We acknowledge this in our manuscript (line 371-374). We conducted preliminary explorations based on existing evidence and our observations, which serve to lay a foundation and provide inspiration for future studies.

Comment 2: The method should be compared to another ROS-limiting effect, such as supplements, which are more easily handled.

Response 2: Thank you for suggesting this research direction, which we find meaningful. It could help compare the effects of different supplements and explore the underlying mechanisms of each supplement's effectiveness. However, our study is in the preliminary phase of exploring the mechanisms of hydrogen gas, and a comparison with a placebo (blank) would better suit our research purpose.

Comment 3: Even if efficacy, inhaling for 1 hour before training or match is very cumbersome.

Response 3: Thank you for considering these aspects. While hydrogen inhalation before routine training can be cumbersome, our research findings are not limited to this scenario. In situations where there is a need to alleviate fatigue or enhance athletic performance, such as in critical competitions or instances of occasional fatigue, hydrogen supplementation presents a scientifically backed and viable intervention option. Similar to pre-exercise warm-ups, hydrogen supplementation has its advantages, and its use is likely to increase as the scientific evidence grows.

Round 2

Reviewer 2 Report

Comments and Suggestions for Authors

Even if the authors tried to answer my questions, they did not substantially change the manuscript.

Comments on the Quality of English Language

Few